# Noncanonical DNA Cleavage by BamHI Endonuclease in Laterally Confined DNA Monolayers Is a Step Function of DNA Density and Sequence

**DOI:** 10.3390/molecules27165262

**Published:** 2022-08-17

**Authors:** Abimbola F. Adedeji Olulana, Dianne Choi, Vincent Inverso, Shiv K. Redhu, Marco Vidonis, Luca Crevatin, Allen W. Nicholson, Matteo Castronovo

**Affiliations:** 1Department of Physics, PhD School in Nanotechnology, University of Trieste, 34127 Trieste, Italy; 2Department of Medical and Biological Sciences, University of Udine, 33100 Udine, Italy; 3Regional Centre for Rare Diseases, University Hospital Udine, 33100 Udine, Italy; 4School of Food Science and Nutrition, University of Leeds, Leeds LS2 9JT, UK; 5Department of Biology, Temple University, Philadelphia, PA 19122-6078, USA; 6Department of Chemistry, University of Rome “Tor Vergata”, 00133 Rome, Italy; 7Department of Life Sciences, University of Trieste, 34127 Trieste, Italy

**Keywords:** DNA, atomic force microscopy, nanografting, self-assembled monolayers, surfaces, endonuclease, noncanonical recognition, nanoscale confinement, DNA nanotechnology

## Abstract

Cleavage of DNA at noncanonical recognition sequences by restriction endonucleases (star activity) in bulk solution can be promoted by global experimental parameters, including enzyme or substrate concentration, temperature, pH, or buffer composition. To study the effect of nanoscale confinement on the noncanonical behaviour of BamHI, which cleaves a single unique sequence of 6 bp, we used AFM nanografting to generate laterally confined DNA monolayers (LCDM) at different densities, either in the form of small patches, several microns in width, or complete monolayers of thiol-modified DNA on a gold surface. We focused on two 44-bp DNAs, each containing a noncanonical BamHI site differing by 2 bp from the cognate recognition sequence. Topographic AFM imaging was used to monitor end-point reactions by measuring the decrease in the LCDM height with respect to the surrounding reference surface. At low DNA densities, BamHI efficiently cleaves only its cognate sequence while at intermediate DNA densities, noncanonical sequence cleavage occurs, and can be controlled in a stepwise (on/off) fashion by varying the DNA density and restriction site sequence. This study shows that endonuclease action on noncanonical sites in confined nanoarchitectures can be modulated by varying local physical parameters, independent of global chemical parameters.

## 1. Introduction

Restriction endonucleases (REases) are bacterial enzymes that carry out sequence-specific cleavage of double-stranded(ds) DNA and are routinely utilised for the manipulation of DNA in diverse applications [1,2]. Each REase specifically recognises a short palindromic sequence (restriction site) within the double helix, and catalyses phosphodiester hydrolysis in each strand, yielding product fragments with overhangs or blunt ends [2]. Noncanonical DNA recognition and cleavage by an REase refers to cleavage at sites that slightly differ in their sequence from the canonical sites. Cleavage at these sites can be promoted by varying parameters, including temperature, pH, or buffer composition, which results in an increase in the number of cuts in the DNA. This is usually an unwanted action of REases (termed “star activity” [3,4,5,6,7]) and has been shown to vary between isoschizomers (REases sharing the same cognate site) depending on structural differences affecting the DNA-binding mechanism [8]. 

The inherent capacity of nucleic acids to self-assemble via canonical Watson–Crick base pairing allows the generation of programmed DNA architectures either on surfaces or in solution [9]. As one example, in chemical nanoreactors, biomolecular components are positioned with nanoscale precision and controlled orientation to locally permit or enhance reaction progression [10,11,12]. This technology is broadly inspired by reactions occurring inside biological cells and may provide novel routes for the multiplexed chemical manipulation of nucleic acids and other biopolymers by directing reactions that normally occur in bulk solution. 

Combining the immobilisation of DNA on solid surfaces using micro and nano-lithographic techniques allows the generation of spatially confined DNA brushes (monolayers) with surface densities that closely approximate that occurring in intracellular microenvironments [11,12,13,14]. By varying the height, composition, orientation, and density of the DNA molecules in such brushes (also termed patches or laterally confined DNA monolayers (LCDMs)), it is possible to investigate biomolecular reactions and interactions to an extent that cannot be achieved in dilute solutions [11,12,15,16]. While other studies have investigated the physical properties of DNA brushes [17,18,19], key applications essentially fall in the area of synthetic biology, including the generation of cell-free gene expression systems [14]. 

Our group used the atomic force microscopy (AFM)-based nanolithography technique termed nanografting [20] to generate LCDMs on ultra-flat gold surfaces [12,13,16,21]. We used such DNA patches to investigate endonuclease action on the immobilised DNA by AFM measurement of the height change in the DNA patch relative to the surrounding inert surface [13,16]. Using the flanking alkylthiol monolayer as a reference, (i) at a constant dsDNA length, the higher the inherent DNA surface density, the higher the dsDNA monolayer; and (ii) at a constant dsDNA density, the longer the dsDNA molecule, the higher the dsDNA monolayer [13,16]. As one example, we found that the accessibility of DpnII REase to its cognate sites is density dependent in an abrupt stepwise manner, with DNA molecules undergoing cleavage at the canonical recognition sequence (positioned in the middle of the DNA) only when a specific DNA density threshold is achieved [16]. Further studies indicated that DpnII and BamHI can access their recognition sites by diffusing within the DNA patch in a two-dimensional fashion, specifically by progressing from the sidewalls of the laterally confined DNA monolayer (about 10 nm in height) rather than from the topmost DNA–liquid interface of several square microns in area [13]. 

In this study, we further investigated the behaviour of BamHI, an endonuclease that recognises a single unique sequence of 6 bp [22,23,24], by focusing on the role of the DNA density in noncanonical recognition. We compared BamHI’s action on the cognate sequence (5-GˆGATCC-3′) with two partially cognate sequence variants, each differing by 2 bp from the cognate sequence. The binding of BamHI to its cognate site and the hydrolysis of the GˆG phosphodiester in each strand have been extensively characterised [22,23,24], in addition to the mechanism of cleavage site recognition and specificity [25,26,27,28]. We used nanografting and AFM topographic imaging to study the enzymatic reaction within laterally confined DNA monolayers (LCDMs), either in the form of small DNA patches or as DNA monolayers on surfaces. Our findings show that at low DNA densities, BamHI recognises only the cognate sequence while at intermediate DNA densities (i.e., with DNA molecules in a near-vertical orientation), noncanonical recognition can occur, and in a stepwise (on/off) fashion as a function of the DNA density and sequence.

## 2. Results 

### 2.1. Design, Fabrication, and AFM Analysis of Surface-Bound, Laterally Confined DNA Monolayers (LCDMs) 

Nanografting is a mechanically assisted nano-lithography technique that uses the AFM tip to transfer mechanical energy onto a selected SAM-coated area on a gold surface. By loading the tip at high forces (~120 nN), the thiol-modified molecules in the SAM are replaced with thiol-modified molecules in the contacting solution [20] (which, in our case, is a thiol-modified ssDNA [13]; see Figure 1a,b). The density of the ssDNA molecules in the nanografted patch can be varied by adjusting the nanografting parameters, as originally developed by Liu [29] and optimised by our group [21]. Essentially, the DNA density is proportional to the number of times the AFM tip over-writes the selected area during the nanografting process, until saturation is achieved [21]. We designed three distinct, 44-nt DNA molecules (termed DNA-1, DNA-2, and DNA-3) with a canonical or noncanonical BamHI recognition sequence located in the middle and generated different LCDMs for each DNA. DNA-1 contains the canonical BamHI recognition site (RS) that supports full reactivity while the other two DNAs contain RS variants that only partially match the cognate RS (RS_DNA-2_: 5′-CGATCA-3′, RS_DNA-3_: 5′-AGATCA-3′) (see Table 1 for the full sequences of DNA-1, DNA-2, and DNA-3). All sequences contain no other 5′-NGATCN-3′ elements except the one in the middle. The LCDMs were generated either via direct DNA nanografting within an ethylene glycol-terminated alkylthiol SAM (direct nanografting approach; see Figure 1) or by spontaneous formation of DNA self-assembled monolayers, followed by nanografting of an ethylene glycol-terminated alkylthiol patch (negative nanografting approach; see Figure 2 and also below).

DNA hybridization was then accomplished by the addition of the complementary ssDNA (Figure 1c). Next, the AFM micrograph of the hybridised LCDM was obtained by imaging the area at low force (~0.2 nN). Notably, the measured topographic height of the dsDNA monolayers directly depends on the DNA density and the length of the DNA duplex. At the maximum DNA density, the dsDNA molecules stand essentially upright in the LCDM [13,20,21], leading to a height saturation value of approximately 14 nm for the 44-bp dsDNA [13]. Successfully prepared LCDMs (80–90% of the total LCDMs prepared) were incubated with 0.2 U/µL (~20 nM) BamHI at 37 °C in the standard reaction buffer (see Methods) for 1 h. After washing the surface, AFM topographic height images were obtained at low force (~0.2 nN), allowing comparison of the height profile of each LCDM before and after the enzymatic reaction (see Figure 1). Since DNA cleavage leads to a double-helical length reduction of 50% for the surface-attached dsDNA (see the diagrams on top in Figure 1e,f), a similar percentage decrease in the LCDM height indicates BamHI catalytic action [13,16].

### 2.2. Noncanonical DNA Cleavage by BamHI within LCDMs

We characterised the ability of BamHI to cleave DNA with different RS within nanografted LCDMs with fixed dimensions of 1 × 1 µm^2^, with approximately the same DNA density, and under the same reaction conditions. The first row of Figure 1 shows the action of BamHI in an LCDM containing the cognate RS (DNA-1), in which the relative BamHI efficiency is maximal. The second and third rows of Figure 1 show the same for non-cognate RS (DNA-2 and DNA-3, respectively), which are the focus of this work. Qualitatively, the “colour” of the LCDM in the AFM micrograph in Figure 1d (before reaction) is changed following the enzymatic reaction (Figure 1g), where the darker colour reflects a topographically shorter structure. The height profiles across the LCDM (see coloured lines in Figure 1d,g) are shown in Figure 1h. In the top row, the initial relative height (H_R_) for DNA-1 (red line) is ~5.5 nm and it is reduced to ~3 nm (blue line) after the enzymatic reaction, corresponding to a ~45% height reduction. The surface density of the DNA-1 molecules can be estimated from the relative height of the LCDM, applying the empirical method introduced in a previous study of our group [13] to an LCDM of 44-bp dsDNA, and providing a value of ~1.6 × 10^3^ molecules/µm^2^. However, subsequent studies suggest that this value should be regarded as a lower limit value of the density [30,31,32]. Similar results are obtained for LCDMs containing DNA-2 or DNA-3 (see the second and third row, respectively, in Figure 1d–h). Thus, noncanonical DNA cleavage by BamHI can occur in an LCDM. Specifically, for DNA-2, H_R_ is initially near 5.5 nm and is reduced to ~4 nm after BamHI action, for a ~23% average height reduction (see the topographic profiles in the middle in Figure 1h). For DNA-3, H_R_ changes from ~6 to ~4 nm after BamHI action, for a ~33% average height reduction (see the topographic profiles at the bottom in Figure 1h). As depicted in the diagrams in the middle and at the bottom of Figure 1f relative to DNA-2 and DNA-3, respectively, an LCDM height reduction of significantly less than 50% reflects a reaction where only a fraction of the dsDNAs in the LCDM are cleaved [13,16].

We applied negative nanografting to further characterise the action of BamHI in LCDMs of varying DNA densities. This approach involves the spontaneous formation of an SAM by the addition of a pre-hybridised, thiol-derivatised dsDNA to a gold surface (see Methods section and Figure 2a,b for details). Compared to direct DNA nanografting, negative nanografting allows the generation of LCDMs with medium to high DNA densities, thereby overcoming the limitations of direct ssDNA nanografting, reflecting poor hybridisation efficiency in dense ssDNA monolayers [21,33] and suboptimal nanografting of 44-bp dsDNA molecules (data not shown). We examined the BamHI reaction in LCDMs of higher density than those presented in Figure 1 (i.e., with H_R_ initially near or slightly more than 10 nm; see DNA-1 and DNA-2, respectively, on the top and at the bottom in Figure 2c–g). By comparing the relative height profiles of the LCDM before and after BamHI action (Figure 2g), it can be seen that DNA cleavage occurs, as evidenced by a height reduction of ~50% in both systems. Of note, for DNA-1 and DNA-3, as H_R_ increases from 5–6 (Figure 1) to near 10 nm (Figure 2), the average percentage height reduction nearly doubles. Therefore, noncanonical DNA cleavage by BamHI is more efficient at higher DNA densities. This result seems counterintuitive, in that, according to what is known for REase star activity in solution, the noncanonical reaction efficiency is generally inversely proportional to the DNA concentration [3]. As such, the hitherto cleavage efficiency in noncanonical LCDM should decrease as a function of the DNA density (see the Discussion). 

### 2.3. Factors Controlling Noncanonical BamHI Action in LCDMs

We examined the DNA density and sequence as factors that may control BamHI’s catalytic action in LCDMs. We utilised both nanolithography approaches as described above to generate, for each DNA sequence, LCDMs exhibiting a wider DNA density range (i.e., LCDM height between 2 and 14 nm). Figure 3 shows additional data with respect to that presented in Figure 1 and Figure 2, and summarises the results obtained with three representative initial H_R_ values (H_I_) for each sequence, corresponding to a low (H_I_ < 5 nm), intermediate (5 nm < H_I_ < 10 nm), and high (H_I_ > 13 nm) DNA density. In LCDMs generated with DNA-1 by positive nanografting at a low (Figure 3a,d) and intermediate (Figure 3b,e) DNA density, BamHI efficiently recognises the canonical sequence, leading to complete DNA cleavage (i.e., 50% height decrease in (g), see the brown and orange dots, respectively) and is consistent with the known efficiency of BamHI’s specificity for its canonical RS in solution [26,28]. The high-density DNA-1 LCDM, generated by negative nanografting (Figure 3c,f), and with an H_I_ > 13 nm, is unaffected by BamHI treatment (see the green dot in Figure 3g). A similar behaviour is observed with high-density LCDMs comprising DNA-2 (Figure 3j,m) and DNA-3 (Figure 3q,t; see also Appendix A). These results indicate that BamHI action is inhibited at high DNA densities (H_I_ > 13 nm), probably through steric hindrance, as also reported for DpnII REase [13,21]. 

The density-dependent behaviour of BamHI in LCDMs containing non-cognate sites (DNA-2 and DNA-3) is more complex (see Figure 3h–n and Figure 3o–u, respectively). Figure 3 clearly shows that noncanonical DNA cleavage by BamHI is fully suppressed for H_I_ < 5 nm (see the height profiles in Figure 3k,r, and the brown dots in Figure 3n,u, for DNA-2 and DNA-3, respectively) but does occur for H_I_ values slightly higher than 5 nm (see the height profiles in Figure 3l,s, and the orange dots in Figure 3n,u, for DNA-2 and DNA-3, respectively). At H_I_ ~7 nm, the reaction is complete (final relative height (H_F_) is ~50% of H_I_) for both directly nanografted LCDMs comprising DNA-2 (see the height profiles in Figure 3l, and the orange dot in Figure 3n) and negatively nanografted LCDMs comprising DNA-3 (see the height profiles in Appendix A, and the orange dot in Appendix A). 

These qualitative results indicate that steric factors, and not merely surface confinement, promote noncanonical BamHI action in a sequence-dependent fashion. Noncanonical BamHI action exhibits a step-like behaviour as a function of the LCDM height (and thus the DNA density), with an activation threshold observed at an H_I_ ~5 nm. In addition, the results show slight topographic differences between directly and negatively nanografted LCDMs, in that the latter exhibits a reduced topographical roughness (see the high-resolution AFM micrographs in Appendix A, which are relative to Figure 1 and Figure 2, and Appendix A). This indicates that the LCDMs generated by direct DNA nanografting comprise a heterogeneous ensemble of subdomains of differing individual densities. If so, the subdomains would differ in reactivity, reflecting the dependence of the DNA cleavage efficiency on the local DNA density. Since the analysis of the data presented in Figure 1, Figure 2 and Figure 3 refers to the LCDM height averages, in the following section, we introduce a different approach to analyse the AFM data that can resolve the efficiency of BamHI’s action as a function of the local DNA density within heterogeneous LCDMs. 

### 2.4. Effect of LCDM Heterogeneity on BamHI Noncanonical Action

For each LCDM, we differentially analysed the AFM height profiles before and after BamHI reaction (see, respectively, the red and blue lines in Figure 4a,b,f,g). After aligning each profile pair (see the red and blue profiles in Figure 4c,h, tallying with the line profiles in Figure 4a,b and Figure 4f,g, respectively), we isolated data points relative to the LCDM (see grey squares in Figure 4c,h), and obtained an H_F_ vs. H_I_ scatter plot for each profile pair (see Figure 4d,i). Finally, single-profile data points were pooled to generate a cumulative H_F_ vs. H_I_ scatter plot for each LCDM micrograph (see Figure 4e,j, left). Figure 4 presents a high-resolution analysis of LCDMs comprising cognate (DNA-1, Figure 4a–e) and non-cognate (DNA-3, Figure 4f–j) DNA at intermediate densities (H_I_ ~5 nm). The graphed data in Figure 4e (left) show that BamHI can homogeneously (i.e., fully) process the dsDNA with the cognate sequence, regardless of local DNA density variation, with a linear fit slope of 0.4 (Figure 4e, right), which is in line with the 50% height reduction described for the DNA-1 LCDM in Figure 2, Figure 3 and Figure 4 (see the dashed lines in Figure 4e, left, corresponding to an H_F_/H_I_ ratio of 100% and 50%, respectively). In contrast, the regression analysis in Figure 4j (left) shows that BamHI exhibits a different behaviour in the non-cognate LCDMs. Here, the slope of the linear fit is 0.9, and despite the H_I_ values exhibiting a very small range (5–7 nm), the H_F_ data points consistently range between 50% and 100% of the initial value (see the dashed lines in Figure 4j, left, which correspond to complete and fully supressed DNA cleavage, respectively). In turn, the lower-density dsDNA monolayer domains (H_I_ ~5 nm) support complete BamHI reaction (near 50% height reduction) while the denser dsDNA monolayer domains (H_I_ ~7 nm) remain substantially unaffected by BamHI, with an average height reduction of ~25%, which is in line with the results illustrated in Figure 3u (see the orange dot). In summary, the high-resolution LCDM reactivity analysis reveals that noncanonical BamHI action, as indirectly reflected by the topographic height, depends on the local DNA density.

The LCDM analysis presented in Figure 4 was extended to a higher DNA density (H_I_ between 6 and 9 nm) (Appendix A). Despite the presence of density inhomogeneity in both cognate (DNA-1) and non-cognate (DNA-3) LCDMs (see Appendix A, respectively), BamHI fully reduces the relative height by ~50% in both LCDMs (see Appendix A, respectively). Cumulatively, the high-resolution differential analyses presented in Figure 4 and Appendix A suggest that BamHI’s dependence on DNA density heterogeneity in non-cognate LCDMs varies as a function of the covered DNA density range while its action is complete in a cognate LCDM, independent of density. 

Encouraged by the high-resolution analysis above, we extended the approach to study BamHI’s behaviour over a broad density range, using both direct and negatively nanografted LCDMs. The results are shown in Figure 5 and Figure 6. In cognate (DNA-1) LCDMs, BamHI action is complete for H_I_ < 14 nm (see Figure 5a), which is a threshold likely reflecting a surface DNA density above which DNA cleavage site accessibility is inhibited by steric hindrance [13]. Surprisingly, noncanonical DNA cleavage by BamHI varies for non-cognate sequences (DNA-2 and DNA-3). In both cases, and in agreement with the qualitative analysis in Figure 3n,u, the DNA cleavage efficiency is an apparent step function of the DNA density, as reflected by the initial height, and presents three consecutive states (OFF, ON, OFF) that are linked by stepwise thresholds. The first threshold is at H_I_ < 5 nm, below which BamHI action is suppressed (see Figure 5b,c). For LCDMs initially higher than 5 nm, the noncanonical BamHI reaction reaches completion only within the lower-density domains of the heterogenous LCDMs. Finally, noncanonical BamHI action is fully inhibited in LCDMs > ~14 nm for DNA-2, > ~11 nm for directly nanografted DNA-3, and ~10 nm for negatively nanografted DNA-3 (see Figure 5b,c and Figure 6).

## 3. Discussion and Conclusions

This study examined the effect of the restriction site sequence and DNA density on the catalytic action of BamHI endonuclease within laterally confined DNA monolayers (LCDMs). We integrated the AFM-based nanolithography method, termed nanografting, with the spontaneous formation of self-assembled monolayers of dsDNA, followed by AFM topographic imaging and analysis. We assessed BamHI action towards three 44-bp dsDNAs that either contained the cognate palindromic recognition sequence or contained noncognate sites that shared the 4-bp palindromic core with the cognate sequence. We found that the nanoscale confinement parameters of the DNA surface arrays determine the ability of BamHI to cleave non-cognate sites. Specifically, (i) a dsDNA monolayer height of ~5 nm is necessary for cleavage of noncognate sites, indicative of a lower-bound DNA density threshold required for reaction; (ii) subdomain density inhomogeneities in the dsDNA monolayers generated by direct DNA nanografting lead to variation in the noncanonical BamHI action, with subdomains of lower density undergoing more efficient cleavage; (iii) BamHI activity is a step function of the DNA height (and therefore density) and sequence, wherein the non-cognate sites differ from one another by a single bp; (iv) BamHI is inhibited at high DNA densities irrespective of the sequence, most likely reflecting steric hindrance as also suggested by previous studies [13,16]; and (v) noncanonical BamHI action is also inhibited at high DNA densities, but the thresholds vary between non-cognate sites, with both being lower than that of the cognate site. 

Therefore, nanoscale confinement appears capable of conferring a novel stepwise noncanonical BamHI catalytic action. It remains unclear, however, what determines the lower-bound DNA density thresholds that control enzyme action. In a previous study, we characterised the diffusion mechanism of BamHI in LCDMs, and provided evidence that the endonuclease is physically retained within the dsDNA monolayer (i.e., beneath the top-most LCDM–solution interface) during reaction [13]. With an average LCDM height greater than the enzyme diameter, the enzyme diffuses within the LCDM architecture in a two-dimensional manner [13]. Therefore, the stepwise behaviour of noncanonical BamHI action would also be connected with 2D diffusion in LCDMs and would be diffusion limited. Indeed, the lower bound threshold of H_I_ ~5 nm could be associated with the 2D diffusion regime, which is compatible with an empirical model linking LCDM height to DNA density [13]. On the other hand, studies of the effect of molecular crowding on biochemical reactions suggest that the DNA molecular density directly impacts enzyme–DNA transactions and is inversely related to enzyme mobility within the highly dense DNA phase (diffusion-limited regime) [34]. In turn, high diffusivity favours enzyme reaction, whereas low diffusivity reduces the probability of enzyme–DNA collision. In turn, these behaviours suggest that DNA density heterogeneity in LCDMs enhances 2D diffusion within lower-density subdomains, in which DNA cleavage would alter (“pierce”) the top-most LCDM–solution interface and disrupt 2D enzyme “trapping”, with the effect of supressing noncanonical BamHI action in the adjacent higher-density domains. Furthermore, the differences in the reactivity of the two non-cognate sites compared to the canonical site may reflect the intrinsically weaker enzyme–DNA-binding interaction. In fact, the GC content deviation from the cognate sequence is greater for DNA-3, which can be correlated to a narrower DNA density window that determines noncanonical BamHI action. In a recent work, Aggarwal and co-workers determined the structure OkrAI, a BamHI isoschizomer, and also showed that OkrAI exhibits higher star activities than BamHI at 37 °C towards bacteriophage lambda DNA in solution, at a high enzyme concentration [8]. The different behaviours can be explained by the structural differences between the two enzymes, in that OkrAI may require a lower activation energy for cleaving dsDNA at star sequences [8]. To our knowledge, however, no reported study on endonuclease action on noncanonical sites under conditions of macromolecular crowding has described a similar stepwise “on/off’ behaviour. In this regard, experimental systems adopted to investigate macromolecular crowding typically comprise dense, disordered polymers in solution, where crowding effects are measured as averages of the micro-state-related behaviours simultaneously occurring in bulk solution [34].

This study shows for the first time that noncanonical endonuclease action can be modulated in confined nanoarchitectures by varying local physical parameters, independent of global parameters such as temperature, pH, or solute concentrations. Overall, our study suggests the need to further characterise the action of nucleic acid processing enzymes in nucleic acid nanostructures on surfaces and in solution. While the noncanonical “star” activity of restriction enzymes acting on DNA in dilute solution is known and typically characterised in that reaction environment, a fuller understanding of emergent steric-dependent enzyme behaviours that are not captured in dilute solution may augment (or enable new) applications of nucleic acid nanotechnology.

## 4. Materials and Methods

### 4.1. Materials

Synthetic C6 thiol-modified single-stranded(ss) DNAs and the corresponding complementary ssDNA sequences were purchased from Integrated DNA Technologies (IDT). The restriction enzyme BamHI-HF was purchased from New England Biolabs (Ipswich, MA, USA) and used with *CutSmart* buffer. C11 alkanethiol (HS-((CH_2_)_11_)-(O-CH_2_-CH_2_)_6_-OH) omega (top) substituted with Top-Oligo-Ethylene Glycol (TOEG 6) was purchased from ProChimia Surfaces Sp. z o.o. (Gdynia, Poland). Sodium chloride, Tris-EDTA, and absolute ethanol (99.8%) were purchased from Sigma Aldrich (Merck KGaA, Darmstadt, Germany). All buffer solutions were prepared using ultra-pure water (milliQ-H_2_O) of 18.2 MΩ·cm resistivity at 25 °C, and filtered before use with a sterile syringe filter (0.22 µm pore size). 

### 4.2. DNA Sequence Design

In total 3 distinct 44-bp-long DNA molecules were designed with a BamHI recognition site (canonical or noncanonical) located in the middle of the sequence. DNA-1 carries the canonical BamHI restriction site (5′-GGATCC-3′) that ensures optimal enzyme reactivity while the other two (DNA-2 and DNA-3) contain sequences that partially match the canonical sequence (5′-CGATCA-3′ and 5′-AGATCA-3′, respectively). The sequences with their corresponding complementary sequences are listed in Table 1. 

**Table 1 molecules-27-05262-t001:** DNA sequences and chemical modifications used in this study.

DNA Molecule	Sequence
DNA-1	SH-((CH)_2_)_6_-5′-CAAAACAGCAGCAATCCAA**GGATCC**GACACCCGATTACAAATGC-3′
DNA-2	SH-((CH)_2_)_6_-5′-CAAAACAGCAGCAATCCAA**CGATCA**GACACCCGATTACAAATGC-3′
DNA-3	SH-((CH)_2_)_6_-5′-CAAAACAGCAGCAATCCAA**AGATCA**GACACCCGATTACAAATGC-3′
	Complementary sequences
cDNA-1	5′-GCATTTGTAATCGGGTGTC**GGATCC**TTGGATTGCTGCTGTTTTG-3′
cDNA-2	5′-GCATTTGTAATCGGGTGTC**TGATCG**TTGGATTGCTGCTGTTTTG-3′
cDNA-3	5′-GCATTTGTAATCGGGTGTC**TGATCT**TTGGATTGCTGCTGTTTTG-3′

### 4.3. Preparation of Ultra-Flat Gold Substrate

Ultra-flat gold substrates were prepared as described [13,21]. Briefly, a sequential deposition of gold on freshly cleaved mica was employed using electron beam evaporation in a vacuum chamber (pressure about 10^−6^ mbar). Gold was deposited at a rate of 0.01 nm s^−1^ until a film of 20 nm was obtained, then the rate of evaporation was increased to 0.1 nm s^−1^ until a 100-nm-thick film was formed on freshly cleaved mica (Mica New York Corp., Deer Park, NY, USA) (clear ruby muscovite) at room temperature. The planar gold sheet of (111) crystallographic plane obtained on mica was sliced into few millimetre squares (approximately 5 × 5 mm^2^) in area using a sharp blade. To form ultra-flat free gold surfaces, the freshly formed gold surface was first bonded to the polished side of a silicon wafer with a drop of SU-8 photoresist adhesive (negative tone photoresist, Micro Resist Technology GmbH, Berlin, Germany). A silicon-gold-mica sandwich was obtained by pressing the polished section of silicon on the gold section of mica. The silicon-gold-mica sandwich squares were cured at 130 °C for at least 24 h. The samples were allowed to cool to room temperature without applying an external cooling system to avoid thermal stresses that can result in the gold film detaching from the mica substrate. Without any further surface treatment, the samples were stored at room temperature, ready to be used (after stripping the mica from the gold surface) for self-assembled monolayer preparation.

### 4.4. Preparation of Top-Oligo-Ethylene-Glycol SAM on Ultra-Flat Gold Substrate

An ultra-flat gold surface was obtained by mechanical stripping of the mica substrate from the silicon-gold-mica sandwich, followed by immersion and incubation in a solution of 100 µM top-oligo-ethylene glycol (TOEG 6, (HS-((CH_2_)_11_)-(O-CH_2_-CH_2_)_6_-OH), Prochimia and Sigma Aldrich) in absolute ethanol, and 1 M NaCl, TE (10-mM Tris-HCl, 1-mM EDTA, pH 7.2 at 25 °C in Milli-Q water) for 6 h. The TOEG SAM provides a bio-repellent monolayer to minimise nonspecific adsorption of biomolecules on the surface. After incubation, serial rinsing was performed in both ethanol and 1 M NaCl, TE(1X) buffer to remove any physically adsorbed contaminants.

## 5. Approach 1: Generation of Laterally Confined DNA Monolayers by ssDNA Nanografting

Nanografting experiments were performed as follows: A TOEG SAM-passivated gold substrate was glued on the home-made liquid cell using an inert polymer (TOPAS). Next, an ssDNA solution (0.1–1 µM ssDNA, 1 M NaCl, TE1X) was evenly dispensed on the SAM and was transferred onto the AFM X-Y scanner stage. Preliminary imaging was performed in tapping mode at low force to obtain a topographic image of the TOEG SAM surface and to select a flat and clean section for nanografting. Afterwards, a 1 × 1 µm^2^ section was selected for the grafting process, and concomitantly at high force (≈100 nN in contact mode), TOEG 6 molecules within the selected 1 × 1 µm^2^ section were substituted with the thiol-modified ssDNA in solution. However, to obtain several patches with different molecular densities, grafting parameters, including the patch dimension, the number of times the AFM tip overwrote a selected area during nanografting, the concentration of thiol-modified ssDNA, and the applied nominal force, were tuned during the nanograting process. After the grafting, a 20 × 20 µm^2^ section that contained the grafted patches was scanned at very low force (high set point) in tapping mode. The domain reference (the section that contained the grafted patches) was saved using SV-align software. After nanografting, the sample was thoroughly rinsed with 1 M NaCl, TE1X buffer (DNA-free buffer). For hybridization, a 1 µM complementary ssDNA solution was evenly dispensed on the surface, and hybridization allowed at 37 °C for 1 h, using a Binder oven (model FD 115, BINDER GmBH, Tuttlingen, Germany). After hybridization, the sample was copiously rinsed with a DNA-free buffer consisting of 1 M NaCl, TE1X. This procedure was applied to the DNA-1, -2, and -3 monolayers.

## 6. AFM Nanografting and Imaging

All AFM experiments (scanning, nanolithography, height detection, and monitoring enzymatic reaction) were carried out with an MFP-3D Classic (also called stand-alone) or an MFP-3D BIO AFM (Asylum Research, Santa Barbara, CA, USA). Nanografting used commercially available pyramidal silicon etched probes, NSC19/no Al and NSC 18/no Al, with a spring constant of 0.6 and 2.8 N/m, respectively (Mikro-Masch, Innovative Solutions Bulgaria Ltd., Sofia, Bulgaria). A soft probe (CSC 38 /no Al, spring constant 0.03 N/m, Mikro-Masch, Innovative Solutions Bulgaria Ltd., Sofia, Bulgaria) was used for imaging the monolayers before and after enzymatic reaction.

## 7. Approach 2: Self-Assembled Double-Stranded DNA Monolayer Formation

Double-stranded DNA SAMs on an ultra-flat gold substrate were formed in a prehybridization reaction by incubating equimolar concentrations of thiol-modified and complementary ssDNA in the presence of 100 mM MgCl_2_ and 1 M NaCl, TE1X buffer (10 mM Tris-HCl, 1 mM EDTA, pH 7.2 at 25 °C in Milli-Q water) at 37 °C for 1 h. Following pre-hybridization, absolute ethanol of a defined volume was added to the DNA solution, followed by addition to ultra-flat gold substrate to allow SAM formation. Different molecular densities were obtained by adjusting the concentration of the pre-hybridised dsDNA, the incubation time for SAM formation, and the volume % of ethanol. After allowing SAM formation, the sample was rinsed twice in DNA-free buffer (1 M NaCl), TE1X (10 mM Tris-HCl, 1-mM EDTA, pH 7.2 at 25 °C in Milli-Q water), then passivated with 100 µM TOEG6 in 1 M NaCl, TE1X buffer for 15 min, and rinsed twice in 1 M NaCl, TE1X buffer after the passivation. The dsDNA SAM was fixed within the home-made liquid cell, covered with 1 M NaCl, TE1X buffer, and transferred onto an AFM X-Y scanner stage for imaging and experimentation. Preliminary imaging was performed in tapping mode at low force to obtain the topographic image of the dsDNA SAM surface, and to identify a flat clean section for nanografting. Nanografting of TOEG6 was performed within the dsDNA SAM in 10-µM TOEG6, 1-M NaCl, and TE1X buffer. The negative nanografting was performed in contact mode using the following parameters: applied load ≈ 100 nN (high set point), point and lines: 256, scan rate: 1 Hz and scan angle 90°, and cantilever probe: NSC 19/no Al.

## 8. AFM Measurements

Tip-sample alignment was accomplished using the SV-alignment software. SV-alignment software superimposes the reference tip-sample image on the video panel and the live tip-sample image. This step alleviates the problem of loss of the experimental domain (where the nanografted patches are located on the sample) and allows continuous experimentation on the same domain. IGOR Pro software version 6.3 (WaveMetrics Inc., Portland, OR, USA) was used for the image processing and average topographic height measurement of the dsDNA patches before and after enzymatic reaction. For the topographic profiles shown in Figure 1, Figure 2 and Figure 3 and Appendix A, a line stripe with a 400-nm width was measured across the cross-section of patches using line profile operation. For the topographic profiles shown in the third row in Figure 4 and Appendix A, and in Appendix A, a line stripe with a 50-nm width was measured across the cross-section of patches using line profile operation. To calculate the relative percentage change in the height of dsDNA after the reaction, the following formula was used:$$Percentage\;change\;\left(\%\right)=\frac{Initial\;height-final\;height}{Initial\;height}\times 100$$

## 9. Restriction Enzyme Reaction

After measurement of the LCDM topographic height, the sample was rinsed with 1 M NaCl, TE1X buffer. To equilibrate the environment for restriction enzyme reaction, the sample was rinsed and incubated with *CutSmart* buffer (New England Biolabs, Ipswich, MA, USA), consisting of 50 mM Potassium Acetate, 20 mM Tris Acetate, 10 mM Magnesium Acetate, and 100 μg/mL BSA pH 7.9 at 25°) for 10 min. The restriction reaction was carried out using 0.2 U/µL enzyme (≈20 nM) at 37 °C for 1 h. The sample was rinsed with *CutSmart* buffer, and then with DNA-free 1 M-NaCl, TE1X buffer prior to imaging.

## Figures and Tables

**Figure 1 molecules-27-05262-f001:**
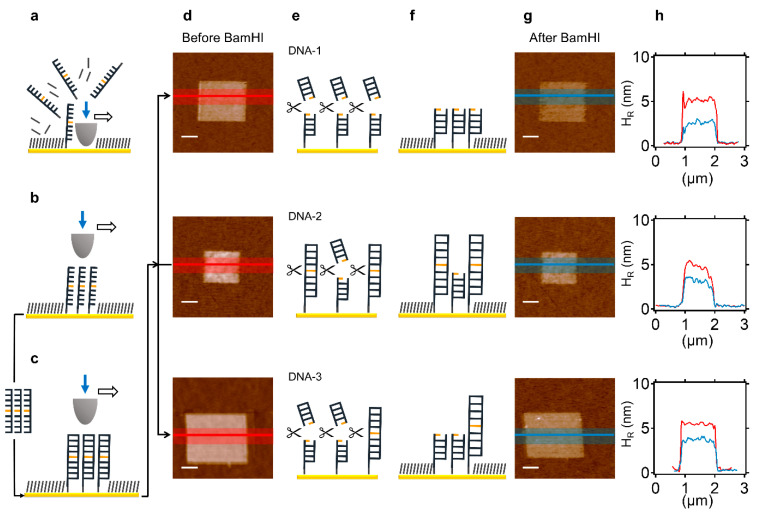
Noncanonical BamHI action in nanografted LCDMs at an intermediate DNA density. (**a**) Nanografting process: at a high applied load (~120 nN), the AFM tip displaces ethylene glycol-terminated alkylthiol molecules from a pre-selected area, consequently allowing immobilization of thiol-modified ssDNA molecules from the contacting solution. (**b**) After washing, a fully complementary ssDNA is added, allowing the formation of dsDNA containing a centrally positioned REase recognition site (depicted in orange). (**c**) Topographic imaging is carried out at a low applied force (~0.2 nN) to register the LCDM height before BamHI reaction, using the surrounding alkylthiol SAM as an inert topographic reference. (**d**) AFM micrograph of an LCDM comprising DNA-1 (top), DNA-2 (middle), or DNA-3 (bottom). (**e**) Schematic representation of the BamHI reaction on an LCDM, creating shortened, surface-bound dsDNAs with high efficiency for DNA-1 (top) or partially efficiency for both DNA-2 (middle) and DNA-3 (bottom) (**f**). (**g**) The same AFM micrograph in (**d**) is imaged after BamHI reaction. (**h**) Average topographic line profiles of the LCDM before ((**d**), red) and after ((**g**), blue) BamHI action, showing a ~45% height decrease for DNA-1 (top), ~23% for DNA-2 (middle), and ~33% for DNA-3 (bottom). The scale bar (white) shown in all the AFM micrographs is 500 nm.

**Figure 2 molecules-27-05262-f002:**
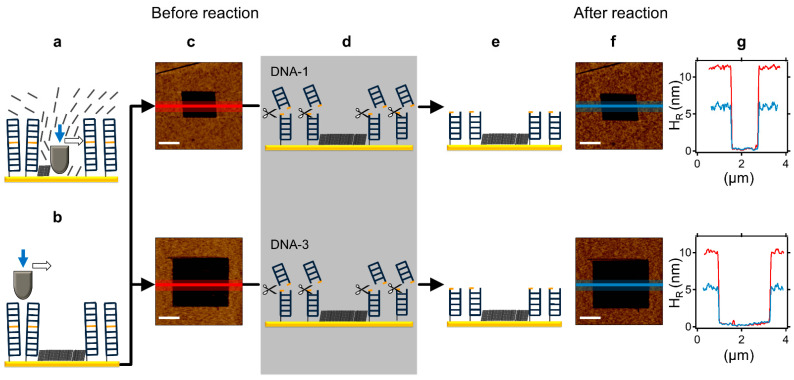
Noncanonical BamHI action in LCDMs exhibiting intermediate DNA density and generated by negative nanografting. (**a**) In negative nanografting, at a high applied load (~120 nN), the AFM tip displaces thiol-modified dsDNA molecules (restriction site indicated in orange) from the pre-selected area, and consequently allows immobilization of ethylene glycol-terminated alkylthiols from the contacting solution. (**b**) After washing, topographic imaging is carried out at a low applied force (~0.2 nN) to register the LCDM height before the BamHI reaction, and using the alkylthiol “pit” as an inert topographic reference. (**c**) AFM micrographs of an LCDM comprising DNA-1 (top) or DNA-3 (bottom). (**d**) Schematic representation of a complete BamHI reaction on an LCDM, creating shorter surface-bound dsDNAs (**e**) and a topographically lower LCDM. (**f**) The same AFM micrographs in (**c**) are imaged after BamHI reaction. (**g**) Average topographic line profiles of the LCDMs before ((**c**), red) and after ((**f**), blue) BamHI action, showing a ~50% height decrease for both DNA-1 (top) and DNA-3 (bottom). The scale bar (white) in all the AFM micrographs is 750 nm.

**Figure 3 molecules-27-05262-f003:**
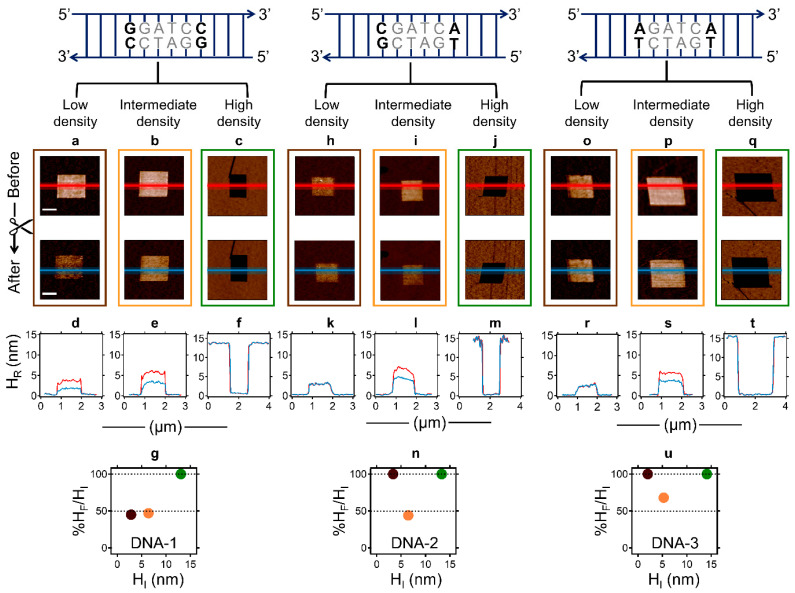
Analysis of noncanonical BamHI action in LCDMs as a function of DNA density and sequence. AFM micrographs in (**a**–**c**,**h**–**j**,**o**–**q**) display LCDMs before (top, red line) and after (below, blue line) BamHI action, for DNA-1, DNA-2, and DNA-3, respectively (see restriction sites above; shared sequence in grey). In each AFM image, the darker area is an ethylene-glycol-terminated alkylthiol monolayer while the coloured line corresponds to the height profiles in (**d**–**f**,**k**–**m**,**r**–**t**), respectively. Representative LCDMs are shown for three DNA densities: low (brown frames in (**a**,**h**,**o**)), intermediate (orange frames in (**b**,**i**,**p**)), and high (green frames in (**c**,**j**,**q**)). Plots in (**g**,**n**,**u**) show the percentage ratio of the relative LCDM height after BamHI action (H_F_) and the initial relative height (H_I_), as a function of three representative H_I_ values for DNA-1, DNA-2, and DNA-3, respectively. The values shown are averages from the height profiles in (**d**–**f**,**k**–**m**,**r**–**t**), respectively. With DNA-1 (**g**), the average height is reduced by half at low (brown; (**a**,**d**)) and intermediate (orange; (**b**,**e**)) densities while at intermediate densities, a similar behaviour is observed for DNA-2 (**i**,**l**,**n**) and DNA-3 (**p**,**s**,**u**). The scale bar (shown in white in panel (**a**)) in all AFM micrographs is 500 nm.

**Figure 4 molecules-27-05262-f004:**
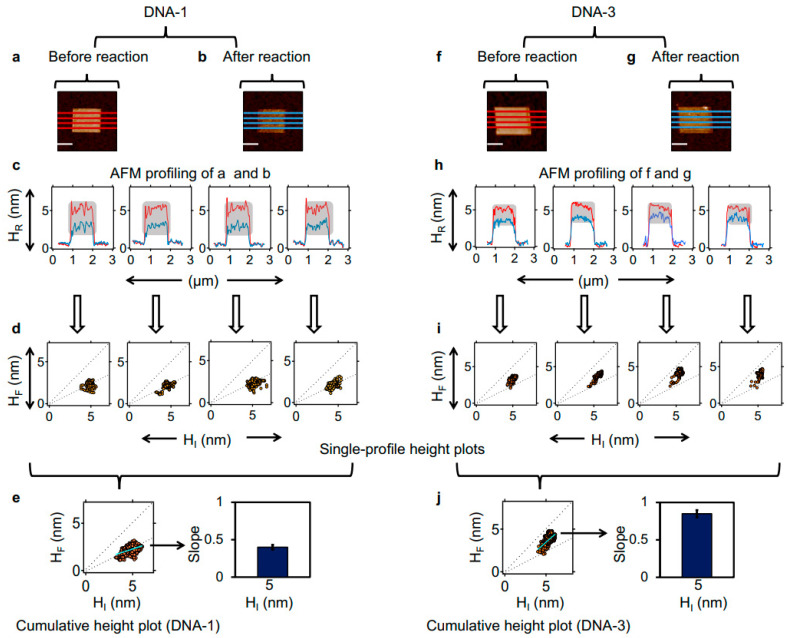
Quantitative differential analysis of BamHI action reveals LCDM density heterogeneity. AFM micrograph of a DNA-1 (cognate) and DNA-3 (non-cognate) LCDM before ((**a**,**f**), respectively) and after BamHI action ((**b**,**g**), respectively). (**c**,**h**): four side-by-side AFM height profiles are extracted from (**a**,**b**,**f**,**g**), respectively. Each pair of AFM profiles (before and after reaction, respectively) correspond to the same nanostructure positions. The H_I_ (red) and H_F_ (blue) data points, exclusively related to the surface-bound DNA in (**c**,**h**) (see grey squares), are filtered and graphed in (**d**,**j**), respectively. The dashed lines are guides corresponding to an H_F_/H_I_ ratio of 100% and 50%, respectively. (**e**,**j**) (left) Data are pooled from (**d**,**i**), respectively; (right) the average slope of the linear fit on the left for LCDMs. In (**a**,**b**,**f**,**g**), the scale bar is 750 nm and the inter-line separation along the slow-scan direction is 200 nm.

**Figure 5 molecules-27-05262-f005:**
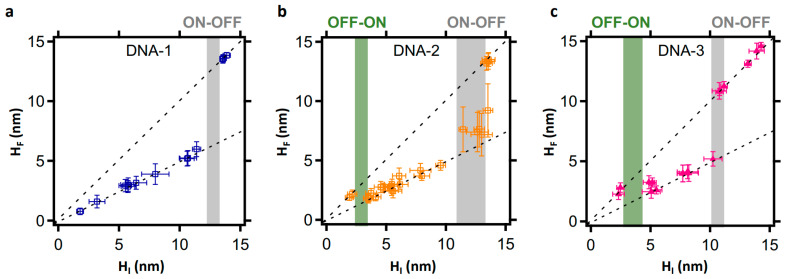
Correlation plots of the relative final height versus the relative initial height, showing the general sequence-density dependence of BamHI action. The plots also show ON-OFF (grey bars) and OFF-ON (green bars) thresholds, indicating the density range where the action of BamHI is ON and the density ranges where it is OFF. The plots in (**a**–**c**) highlight the behaviour of BamHI in in LCDMs, made up of DNA-1, DNA-2, and DNA-3, respectively.

**Figure 6 molecules-27-05262-f006:**
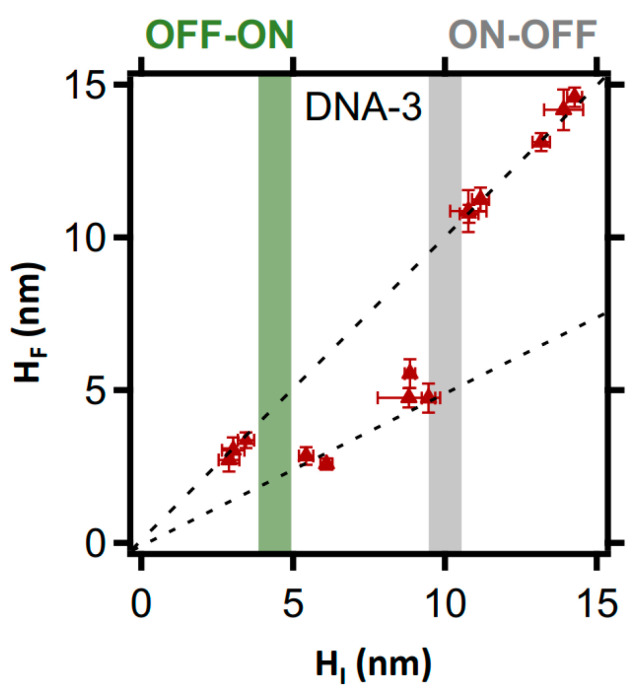
Correlation plot of the relative final height versus the relative initial height, showing the unexpected action of BamHI within DNA-3 LCDMs as revealed by side-by-side AFM height measurements. The plot also shows an OFF-ON (green bar) and an ON-OFF (grey bar) thresholds, indicating the density range where the action of BamHI is ON and the density ranges where it is OFF.

## Data Availability

Raw data are available on request.

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
