# Peer review of "Noncanonical DNA Cleavage by BamHI Endonuclease in Laterally Confined DNA Monolayers Is a Step Function of DNA Density and Sequence"

_molecules, 2022, doi:10.3390/molecules27165262_

Round 1

Reviewer 1 Report

The paper studies the influence of DNA sequence, density and length on the efficiency of the restriction enzyme BamHI-HF . By a clever approach the authors prepare different patches of dsDNA attached to the surface, and vary the sequence for one containing the recognition sequence for the chosen enzyme, or a slightly changed one. After the preparation, which is by AFM-based nanografting either directly (DNA is attached where the AFM tip removes SAM molecules) or indirectly (in a DNA layer, the tip removes DNA molecules which are replaced by SAM molecules), the height of the patches is measured by AFM before and after enzyme incubation. It is a clever experiment with interesting result, and certainly worth publishing.

However, it needs some editing, especially to remove many repetitions in the figure captions, in order to make it better readable.

In Fig. 1, AFM cross-sestions are used to compare the structures before and after nuclease treatment, and to demonstrate the action of nuclease. The sections are quite noisy. I suggest to check if an averaged cross section (not along a line, but along a stripe, averaging about an area) would be less noisy and more convincing? If I remember correctly there was a mode called “line averaging” in the cross sectioning options of AFM software. The same applies to the cross-sections in fig. 2 and 3. As the roughness is later discussed in the paper, these cross-sections as shown there should be added, too, but e.g. in the supplementary information.

In Fig. 1, the figure caption is rather a complete paragraph: All three cases are explained in all details. Thereby, many repetitions result in the long text. It took me quite some time to extract the important information (the numbers for the height differences in h, m, r. I   I would prefer to combine the three description of similar steps (like e.g. d/i/n) in one, so significantly shortening and condensing the caption.  

In figure 2, d and I seems to mislead the reader: In both cases, the depression in the DNA layer made by nanografting is not visible, so I suggest to include a gap in the center (as done in e and j).

I am not fully understanding the relationship between figs. 1 and 3. To me, fig. 1 contains a subset of informations also shown in fig. 3. If this is not the case, please explain more clearly.

I looked for the information how the sectioning was conducted (and the percentage of height reduction determined) but could not find it. Please add or make it clearer.

Line 215/6

We examined the role of DNA density and DNA sequence in as factors that may 216 control BamHI catalytic action in LCDMs. 

Please check the sentence, especially “in as factor”. Probably “in” can be removed here?.

Author Response

The paper studies the influence of DNA sequence, density and length on the efficiency of the restriction enzyme BamHI-HF. By a clever approach the authors prepare different patches of dsDNA attached to the surface, and vary the sequence for one containing the recognition sequence for the chosen enzyme, or a slightly changed one. After the preparation, which is by AFM- based nanografting either directly (DNA is attached where the AFM tip removes SAM molecules) or indirectly (in a DNA layer, the tip removes DNA molecules which are replaced by SAM molecules), the height of the patches is measured by AFM before and after enzyme incubation. It is a clever experiment with interesting result, and certainly worth publishing.

However, it needs some editing, especially to remove many repetitions in the figure captions, in order to make it better readable.

Response:

We thank the reviewer for their comments and suggestions, which have allowed us to improve this manuscript.

- In Fig. 1, AFM cross-sections are used to compare the structures before and after nuclease treatment, and to demonstrate the action of nuclease. The sections are quite noisy. I suggest to check if an averaged cross section (not along a line, but along a stripe, averaging about an area) would be less noisy and more convincing? If I remember correctly there was a mode called “line averaging” in the cross sectioning options of AFM software. The same applies to the cross-sections in fig. 2 and 3. As the roughness is later discussed in the paper, these cross-sections as shown there should be added, too, but e.g. in the supplementary information.

Response:

We agree with the reviewer that the AFM single-line profiles shown in Figs 1-3 were noisy. We have now replaced the previous profiles with others obtained by cross section averaging over a stripe of 400 nm in width. An explanation of the averaging procedure has been added in the methods, under “AFM measurements”. Following the reviewer’s suggestion, we have created a new supplementary figure (now S1) to show the more noisy topographic profiles formerly shown in Figs 1-2.

- In Fig. 1, the figure caption is rather a complete paragraph: All three cases are explained in all details. Thereby, many repetitions result in the long text. It took me quite some time to extract the important information (the numbers for the height differences in h, m, r. I I would prefer to combine the three description of similar steps (like e.g. d/i/n) in one, so significantly shortening and condensing the caption.

Response:

We agree with the reviewer that Fig. 1 caption was too long, presenting redundant information. We have followed the reviewer’s suggestion of reducing the information presented therein and simplifying the numbering in Fig. 1, by combining the description of similar steps. For consistency, we have applied the same revision approach to Fig. 2, whose caption also has been reduced.

- In figure 2, d and I seems to mislead the reader: In both cases, the depression in the DNA layer made by nanografting is not visible, so I suggest to include a gap in the center (as done in e and j).

Response:

We have now modified Fig 2, with all diagrams explicitly presenting the depression in the DNA layer made by nanografting.

- I am not fully understanding the relationship between figs. 1 and 3. To me, fig. 1 contains a subset of informations also shown in fig. 3. If this is not the case, please explain more clearly.

Response:

The AFM micrographs presented in Fig 3 are additional with respect to those shown in Figs 1-2. We have now clarified this aspect in the body text in the 4th line at page 7 of the revised manuscript.

- I looked for the information how the sectioning was conducted (and the percentage of height reduction determined) but could not find it. Please add or make it clearer.

Response:

We have now added information relative to sectioning and calculation of percentage height reduction in the methods, under “AFM measurements”.

- Line 215/6: “We examined the role of DNA density and DNA sequence in as factors that may 216 control BamHI catalytic action in LCDMs. ”. Please check the sentence, especially “in as factor”. Probably “in” can be removed here?

Response:

The typo has been removed. Furthermore, we have extensively proof-read the manuscript again and removed several typos that we had initially overlooked.

Reviewer 2 Report

In this study, the authors have examined the effect of restriction site sequence and DNA density on the catalytic action of BamHI endonuclease within laterally confined DNA monolayers (LCDMs). They integrated the AFM-based nanolithography method termed nanografting along with the spontaneous formation of self-assembled monolayers of dsDNA, followed by AFM topographic imaging and analysis. Authors assessed BamHI action towards three 44bp dsDNAs, that either contained the cognate palindromic site or contained non cognate sites that shared the 4 bp palindromic core with the cognate sequence. Abimbola F. et al. found that nanoscale confinement parameters of the DNA surface arrays can determine the ability of BamHI to process non-cognate sites. Specifically, (i) a dsDNA monolayer thickness of ~5 nm is necessary for cleavage of noncognate sites, indicative of a lower-bound DNAdensity threshold required for reaction; (ii) subdomain density inhomogeneities in dsDNA monolayers generated by direct DNA nanografting lead to variation in off-target BamHI action, with subdomains of lower density undergoing more efficient cleavage; (iii) BamHI activity is a step function of DNA height (and therefore density) and restriction site sequence wherein the non-cognate sites differ by a single bp; (iv) BamHI is inhibited at high DNA densities irrespective of sequence, most likely reflecting steric hindrance as suggested by previous studies (12, 15); (v) off-target BamHI action also is inhibited at high DNA densities, but the thresholds found vary between non-cognate sites, both being lower than that of the cognate site.

This study shows that off-target endonuclease action can be modulated in confined nanoarchitectures by varying local physical parameters, independent of global parameters such as temperature, pH, and solute concentrations. Abimbola F. et al. suggest the need to further characterize the action of nucleic acid processing enzymes in nucleic acid nanostructures on surfaces as well as in a solution. Emergent, steric-dependent enzyme behaviors that are not captured in dilute solution may lead to new biotechnological applications of nucleic acid nanotechnology.

Nanolithography and nanografting have a wide range of applications and these are well-established methods for biosensing, optoelectronics and single molecules manipulation. In contrast, BamHI binds at the recognition sequence 5'-GGATCC-3' , and cleaves these sequences just after the 5'-guanine on each strand. This cleavage results in "sticky ends" which are 4 b.p. Reviewer don’t believe that AFM-based nanolithography and nanografting is an appropriate wetlab method to prove the ff-target DNA cleavage site by BamHI. Authors should do their homework on the Identification of off-target cleavage sites and include more supporting data to establish their hypothesis.

Author Response

Reviewer don’t believe that AFM-based nanolithography and nanografting is an appropriate wetlab method to prove the ff-target DNA cleavage site by BamHI. Authors should do their homework on the Identification of off- target cleavage sites and include more supporting data to establish their hypothesis.

Response:

We thank the reviewer for their comments and suggestions, which have allowed us to improve this manuscript. Following their suggestion, we have expanded the introduction and added a reference (new 8) that suggests that the strength of REase-DNA binding may have a role in REase star activity, as a function of enzyme concentration. Accordingly, we have changed the first paragraph of the introduction and expanded the discussion section at the bottom of page 12 of the revised manuscript to support our hypothesis that DNA density underpins noncanonical activity of BamHI on surfaces. Furthermore, in the conclusive paragraph of the discussion, we state that star activity is typically investigated in dilute solutions (not on surfaces). Since our data show that BamHI can cleave DNA on surfaces at noncanonical sites (not at any off-target sequence, as pointed out by the reviewer), in the first paragraph of the revised results section we have clarified that all DNA molecules involved in this study contain a single 5’-NGATCN-3’ noncanonical site in the middle of the molecule. Furthermore, along this line, we have modified our manuscript, including the title, by replacing the term “off-target” with “noncanonical”, which we believe is more appropriate to describe our findings.

Round 2

Reviewer 2 Report

Besides AFM authors didn't provide any other supporting experimental evidence to prove their hypothesis which was mentioned in the 1st review. I hope the authors understand the meaning of review and comments.